# The Hippo signalling pathway coordinates organ growth and limits developmental variability by controlling *dilp8* expression

Emilie Boone[1], Julien Colombani[1], Ditte S. Andersen[1] & Pierre Léopold[1]

Coordination of organ growth during development is required to generate fit individuals with fixed proportions. We recently identified *Drosophila* Dilp8 as a key hormone in coupling organ growth with animal maturation. In addition, *dilp8* mutant flies exhibit elevated fluctuating asymmetry (FA) demonstrating a function for Dilp8 in ensuring developmental stability. The signals regulating Dilp8 activity during normal development are not yet known. Here, we show that the transcriptional co-activators of the Hippo (Hpo) pathway, Yorkie (Yki, YAP/TAZ) and its DNA-binding partner Scalloped (Sd), directly regulate *dilp8* expression through a Hpo-responsive element (HRE) in the *dilp8* promoter. We further demonstrate that mutation of the HRE by genome-editing results in animals with increased FA, thereby mimicking full *dilp8* loss of function. Therefore, our results indicate that growth coordination of organs is connected to their growth status through a feedback loop involving Hpo and Dilp8 signalling pathways.

[1] University Nice Sophia Antipolis, CNRS, Inserm, iBV, Nice 06100, France. Correspondence and requests for materials should be addressed to J.C. (email: julien.colombani@unice.fr) or to P.L. (email: leopold@unice.fr).

The classic reciprocal grafting experiments of Twitty and Schwind[1] on salamanders have established >80 years ago that organs follow autonomous growth programs. The principle of autonomous organ growth was later confirmed in various animals including insects. When young larval imaginal discs are transplanted in heterologous environments, they reach final sizes that are comparable to those achieved *in situ*[2]. To achieve individuals of correct size and proportions, mechanisms allowing organ growth to be coordinated and coupled with the developmental programme are essential. Evidence that such coordination mechanisms exist comes from studies showing that artificially slowing down growth of a subset of discs delays metamorphosis and reduces the growth rate of unperturbed discs, allowing perturbed discs to complete their growth programs before entry into metamorphosis[3–5]. We and others recently identified the hormone Dilp8 as a central organizer of organ growth coordination[6,7]. Dilp8 is secreted from abnormally growing organs and acts remotely on the central brain complex to delay entry into metamorphosis[6,7]. Dilp8 signals through the Leucine-rich repeat-containing G-protein-coupled receptor 3 (Lgr3) in a pair of bilateral brain (GCL) neurons activating a neuroendocrine circuit that ultimately suppresses synthesis of the moulting hormone ecdysone[8–11]. Interestingly, mutations in the *dilp8* and *lgr3* loci produce animals that exhibit FA. FA, measured as the variance between left and right bilateral traits within individuals, is an assessment of stochastic developmental variations[12]. Therefore, the *dilp8/lgr3* axis carries additional function in adjusting organ size and ensuring developmental stability[7–9]. Reducing *lgr3* levels in the GCL neurons recapitulates the *lgr3* mutant phenotype consistent with organ growth being adjusted through a central relay[8]. The ability of *dilp8* to fine-tune organ growth suggests that its expression should be controlled by signals central to organ size assessment mechanisms.

In this study, to gain insight into how *dilp8* expression might be coupled with organ growth, we searched for regulators of *dilp8* expression among 120 candidates recovered in a genetic screen for molecules coupling disc growth with developmental transitions[6]. The condition used for this screen corresponds to a disc-specific knockdown of the syntaxin Avalanche (Avl; *rn > avl* RNAi), which generates neoplastic growth and a Dilp8-dependent delay in larva-to-pupa transition. We inferred that reducing the function of molecules regulating *dilp8* expression should rescue the delay of *rn > avl* RNAi animals. Indeed, altering JNK signalling efficiently rescues the delay by suppressing the upregulation of *dilp8* transcription observed in *rn > avl* RNAi animals[6]. JNK signalling induces *dilp8* transcription in response to various stresses, including wound healing and tumour formation. This likely represents an important checkpoint mechanism allowing the organism to recover before entering metamorphosis, but may not be important for coordinating organ growth in normally developing animals[6].

## Results

### *dilp8* expression requires the co-activators Yorkie and Scalloped.
In addition to JNK signalling, we identified the Hpo pathway as an important regulator of *dilp8* expression. The Hpo pathway is an important regulator of organ growth and is thought to play a central role in organ size assessment[13,14]. The core kinase module of the Hpo pathway includes the Hpo (Mst1/2 in humans) and Warts/Lats kinases, which suppress activation of the transcriptional co-activator Yorkie (Yki; YAP/TAZ in humans). When the Hpo pathway is inactive, Yki and its DNA-binding partner Scalloped (Sd) activate target genes and promote organ growth[15–20]. We observed that reducing levels of

the transcriptional co-activators Yki or Sd efficiently rescues the developmental delay in *rn > avl* RNAi animals and normalizes *dilp8* transcript levels (Fig. 1a,b). Given the substantial evidence that crosstalk takes place between the Hpo and JNK signalling pathways[21,22], we tested whether Yki can regulate *dilp8* expression independently of JNK signalling. Indeed, we found that *dilp8* expression is still significantly upregulated by Yki overexpression in flies that are mutant for the JNK kinase Hemipterous (Hep; Fig. 1c–i). We next tested whether overexpression of *yki* is sufficient to activate *dilp8* transcription. Using an enhanced green fluorescent protein trap inserted in the first intron of the *dilp8* gene as a reporter for native *dilp8* expression, we could observe increased levels of enhanced green fluorescent protein in *yki*-overexpressing clones (Fig. 2a,b). Consistent with this, Dilp8 protein levels were also elevated in these clones (Fig. 2c,d). In agreement with Yki-regulating gene expression through association with its co-activator Sd, we found that reducing Sd levels in Yki-overexpressing clones abolished the Yki-dependent upregulation of Dilp8 (Fig. 2e–j). We next examined *dilp8* levels in clones carrying mutations in genes encoding upstream components of the Hpo pathway including *expanded* (*ex*), *hpo* and *warts* (*wts*). As expected, reducing the activity of upstream Hpo pathway members, which induces Yki activation, also increases Dilp8 protein levels (Fig. 2k–p).

### A Hippo-responsive element controls *dilp8* expression by Yki/Sd.
Genome-wide CHIPseq analyses using anti-Yki antibodies recently identified a number of potential Yki target genes[17,23]. Interestingly, these CHIPseq data identified a 600-bp promoter fragment localized 1.5 kb upstream of the coding region in the *dilp8* locus. Close examination of this fragment reveals three potential Sd-binding sites (hereafter referred to as Hpo-responsive element (HRE))[23], suggesting that *dilp8* expression might be directly activated by a Yki/Sd heterodimer (see region map in Fig. 3a). To directly test this, we performed DNA pull-down assays by mixing lysates from cells expressing tagged Sd (Sd-Flag) and a 600-bp DNA fragment of the *dilp8* promoter region centred around the HRE (*dilp8* promoter fragment: *dilp8-PF*). A region in the *diap* promoter that is known to bind Sd (*diap-PF*) was used as positive control. We found that Sd-Flag binds the *dilp8-PF* and *diap-PF* with similar efficiency (Fig. 3b). Moreover, the binding of Sd-Flag to the *dilp8-PF* is abolished on targeted mutation of the three putative Sd-binding sites (*dilp8-PFΔ123*; Fig. 3b, Supplementary Fig. 1 and see the 'Methods' section). To test the functional relevance of this binding assay, we analysed the potential of each of these fragments to promote transcription of the *luciferase* reporter gene in the presence of Sd and Yki. Consistent with the DNA pull-down result, both *diap-PF* and *dilp8-PF*, but not *dilp8-PFΔ123*, were able to activate transcription in the presence of Yki and Sd (Fig. 3c). To study the transcriptional regulation of *dilp8* by Yki *in vivo*, we next generated transgenic fly lines carrying constructs harbouring either the full *dilp8* promoter (*dilp8-full-prom*), *dilp8-PF*, *dilp8-PFΔ123* or intron 1 of the *dilp8* gene used as a negative control, all fused to the *lacZ* coding sequence (Fig. 3a). We found that both *dilp8-full-prom* and *dilp8-PF* were able to promote *lacZ* expression in Yki-overexpressing clones (Fig. 3d–o). Moreover, the ability of Yki to induce *lacZ* expression depends on the integrity of the HRE, since mutation of the three Sd-binding sites prevented all *lacZ* expression (Fig. 3p–s). Altogether these results are consistent with Yki-activating *dilp8* gene expression through Sd bound to the HRE in the *dilp8* promoter.

### Yki/Sd ensures developmental stability through *dilp8* expression.
We ultimately wanted to know whether the Yki-dependent regulation of Dilp8 function is central to the mechanism allowing

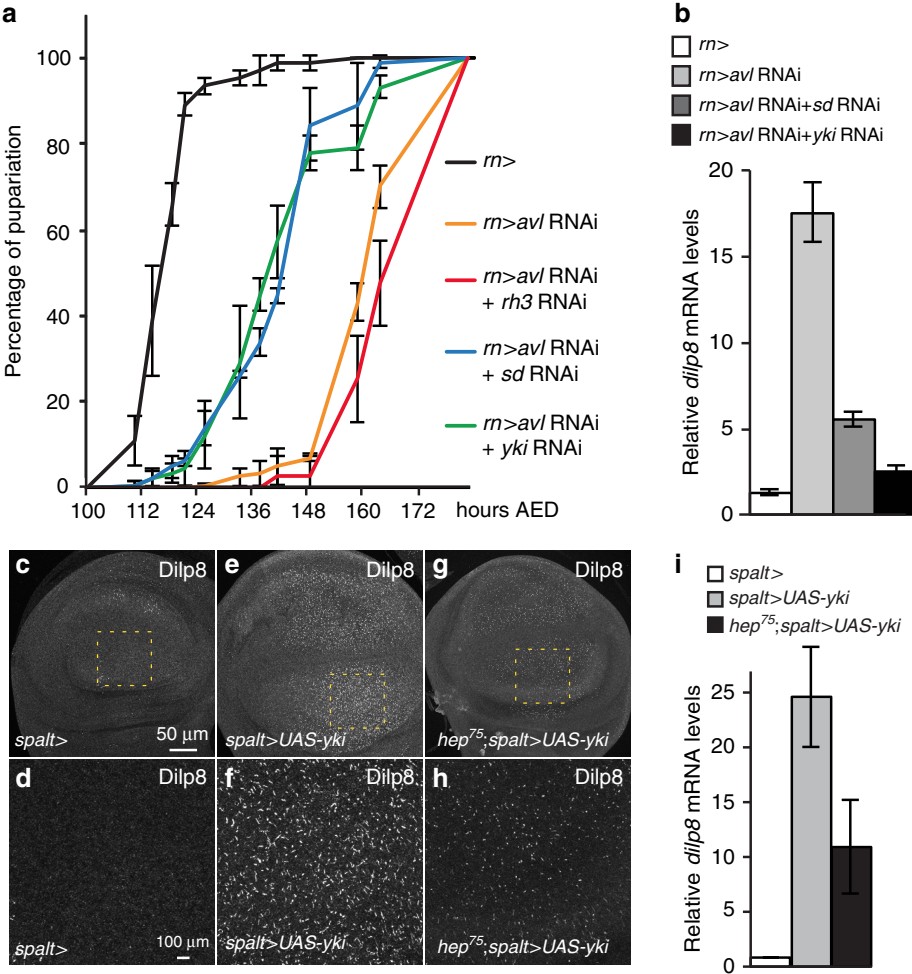

**Figure 1 | Yki regulates *dilp8* expression independently of JNK signalling.** (**a**) Reducing Yki or Sd levels rescues the delay of $rn > avl$ RNAi animals. The percentage of larvae that have pupariated at the indicated hours AED is shown ($n > 90$, triplicate experiments, error bars represent s.e.m.). (**b**) Silencing of Yki or Sd efficiently suppresses the upregulation of *dilp8* mRNA levels observed in $rn > avl$ RNAi animals. Relative *dilp8* mRNA levels were measured using qPCR on whole larvae of the indicated genotypes (triplicate experiments, error bars represent s.e.m.). (**c,e,g**) Yki induces *dilp8* expression independent of JNK signalling. (**d,f,h**) represent higher magnification images of the area shown in the dashed lines. Larval wing discs of the indicated genotypes were dissected 116 h AED and stained for Dilp8. (**i**) qPCRs on the indicated genotypes showing that the Yki-dependent induction of *dilp8* is still significant in a $hep^{75}$ mutant background (triplicate experiments, error bars represent s.e.m.).

organ growth coordination. To test the biological relevance of the HRE in the *dilp8* promoter, we carried out gene editing using a combination of CRISPR-induced double-strand breaks and ends-out homologous recombination[24] to replace the endogenous *dilp8-PF* with either *dilp8-PFΔ123* or a control *dilp8-PF* at the *dilp8* locus (see Supplementary Figs 2 and 3 and see the 'Methods' section). As expected, Yki-dependent induction of *dilp8* expression was efficiently suppressed in flies carrying mutations in the HRE (*dilp8-PFΔ123*; Fig. 4a–d). By contrast, JNK-dependent regulation of *dilp8* expression was not compromised in this genetic background (Fig. 4e–h). We observed that *dilp8-PFΔ123* flies exhibit increased levels of FA, although flies carrying a replacement with the control *dilp8-PF* do not (Fig. 4i, Supplementary Fig. 4). Finally, wing size distribution is neither modified in *dilp8-PFΔ123* flies, nor in *dilp8KO/KO* flies compared with control animals (Supplementary Fig. 5). This suggests that the FA phenotype is not due to a disruption of a general size-control mechanism operating independently in each disc. Overall, our data demonstrate that Yki-dependent regulation of *dilp8* expression plays a critical role in limiting developmental variability. It also contrasts with our observation that *hep75* mutant flies do not exhibit elevated FA (Supplementary Fig. 6). These results are consistent with a

central role of Yki in coordinating organ growth and limiting developmental variability through its effect on *dilp8* expression.

## Discussion

On the basis of a flurry of experiments, different working models have been proposed to explain organ-autonomous size control operating in complex organisms. One unifying view is that the Hpo pathway plays an instrumental role in assessing and regulating organ growth. The ability of the Hpo pathway to integrate both short- and long-range signals makes it an ideal candidate for sensing organ size and regulate growth accordingly. Importantly, we demonstrate here, direct molecular and functional links between Hpo/Yki signalling and *dilp8* expression during normal development. Furthermore, suppressing the sole input of Yki on *dilp8* expression suffices to increase developmental instability and recapitulates the effect of a *dilp8* loss-of-function mutation. The coupling between Hpo pathway activity and *dilp8* expression allows the growth status of organs to be transduced into a Dilp8 signal. Dilp8 acts through its receptor Lgr3, residing in two pairs of bilateral neurons to control the levels of the steroid hormone ecdysone. This in turn affects

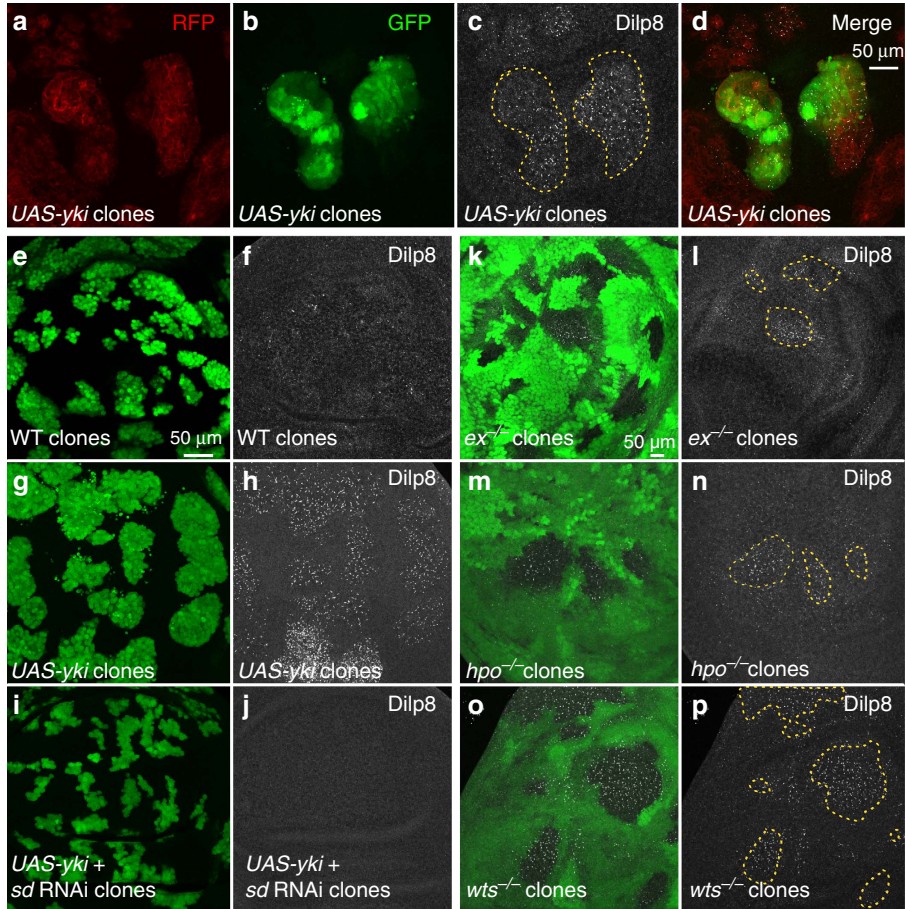

**Figure 2 | Yki regulates *dilp8* transcription through Sd.** Wing imaginal disc carrying RFP-labelled *yki*-expressing clones (**a–d**), GFP-labelled *wt* clones (**e,f**), GFP-labelled *yki*-expressing clones (**g,h**), GFP-labelled *yki* + *sd* RNAi clones (**i,j**), or GFP-negative clones mutant for *ex* (**k,l**), *hpo* (**m,n**) or *wts* (**o,p**), all dissected 116 h AED and stained for Dilp8 (**c,f,h,j,l,n,p**). In **b**, *dilp8* expression is followed by GFP using the *dilp8-GFP* insertion allele (see the 'Genotypes' section under the 'Methods' section).

the timing of maturation, allowing for a checkpoint mechanism on the developmental transition[8–10]. When ectopically expressed, Dilp8 also affects organ growth[6], suggesting that it may also participate in a continuous feedback ensuring proper organ growth during development. Whether ecdysone acts downstream of Dilp8/Lgr3 to mediate such continuous feedback on organ growth remains to be determined. Several reports indicate that ecdysone is required for imaginal tissue growth both in flies and in Lepidoptera[4,5,25–26], and previous data suggest that organ growth coordination relies on systemic effects mediated by ecdysone. Interestingly, Taiman, a co-activator of ecdysone receptor mediates Yki-dependent *dilp8* expression, therefore suggesting that the ecdysone signal itself feeds back on *dilp8* expression[27].

On the basis of previous experimental data, two non-exclusive models can be proposed for organ growth coordination during development. In a developmental checkpoint model, each tissue autonomously follows its growth trajectory and deviations that may occur due to developmental noise are resolved at the end of the growth period by adjusting the time of the next developmental transition. Alternatively, continuous adaptation of growth rates and/or organ size could take place during development, allowing organs to adjust while growing. Although experimental evidence exists for both mechanisms in response to large growth disruptions, it is not clear whether they apply for small perturbations usually encountered by larval organs during development. Further analysis will be necessary to decipher how

the Yki/Dilp8 coupling contributes to final organ size adjustment and limits developmental noise.

## Methods

**Fly strains and food.** The following RNAi lines were from the GD or KK collections of the Vienna *Drosophila* RNAi Center (VDRC): *yki* RNAi (KK104523), *sd* RNAi (KK108877), *avl* (GD107264) and *rh3* RNAi (KK100853). The *UAS-yki* (ref. 15); FRT82B, *LATS*[X1] (ref. 28); FRT42D, *hpo*[5-1] (ref. 29); FRT82B, *ex*[e1] (ref. 30) lines were provided by Nic Tapon (The Francis Crick Institute, London, UK). The *elav-Gal80* was kindly provided by Alex Gould (The Francis Crick Institute). The *yw*, *hep*[75]/FM7, GFP; Sal[PE]/Cyo (ref. 31) line was kindly provided by Florenci Serras (University of Barcelona, Barcelona, Spain). The *UAS-hep*[CA]; rn-Gal4; *dilp8-GFP* and other lines were provided by the Bloomington Drosophila Stock Center.

Animals were reared at 25 °C (or 26.5 °C for Fai measurement) on fly food containing, per litre: 17 g inactivated yeast powder, 83 g corn flour, 10 g agar, 60 g white sugar and 4.6 g Nipagin.

**Cell culture.** *Drosophila* S2R$^+$ cells (DGRC) were grown in Complete Schneiders medium (Schneider medium (Invitrogen) supplemented with 10% heat-inactivated fetal bovine serum (BioWhittaker), 50 U ml$^{-1}$ penicillin, and 50 µg ml$^{-1}$ streptomycin (Invitrogen)) at 25 °C. Transfections were done using Effectene reagent (Qiagen).

**Immunostainings of larval tissues.** Tissues dissected from larvae in 1 × phosphate-buffered saline (PBS (137 mM NaCl, 2.7 mM KCl, 4.3 mM Na$_2$HPO$_4$, 1.47 mM KH$_2$PO$_4$, pH 8)) at the indicated hours after egg deposition (AED) were fixed in 4% formaldehyde (Sigma) in PBS for 20 min at room temperature, washed in PBS containing 0.1% Triton-X-100 (PBT), blocked for 2 h in PBT containing 10% fetal bovine serum (PBS-TF) and incubated overnight with primary antibodies at 4 °C. The next day, tissues were washed, blocked in PBS-TF and incubated with secondary antibodies at 1/500 dilution (Cy3-conjugated donkey

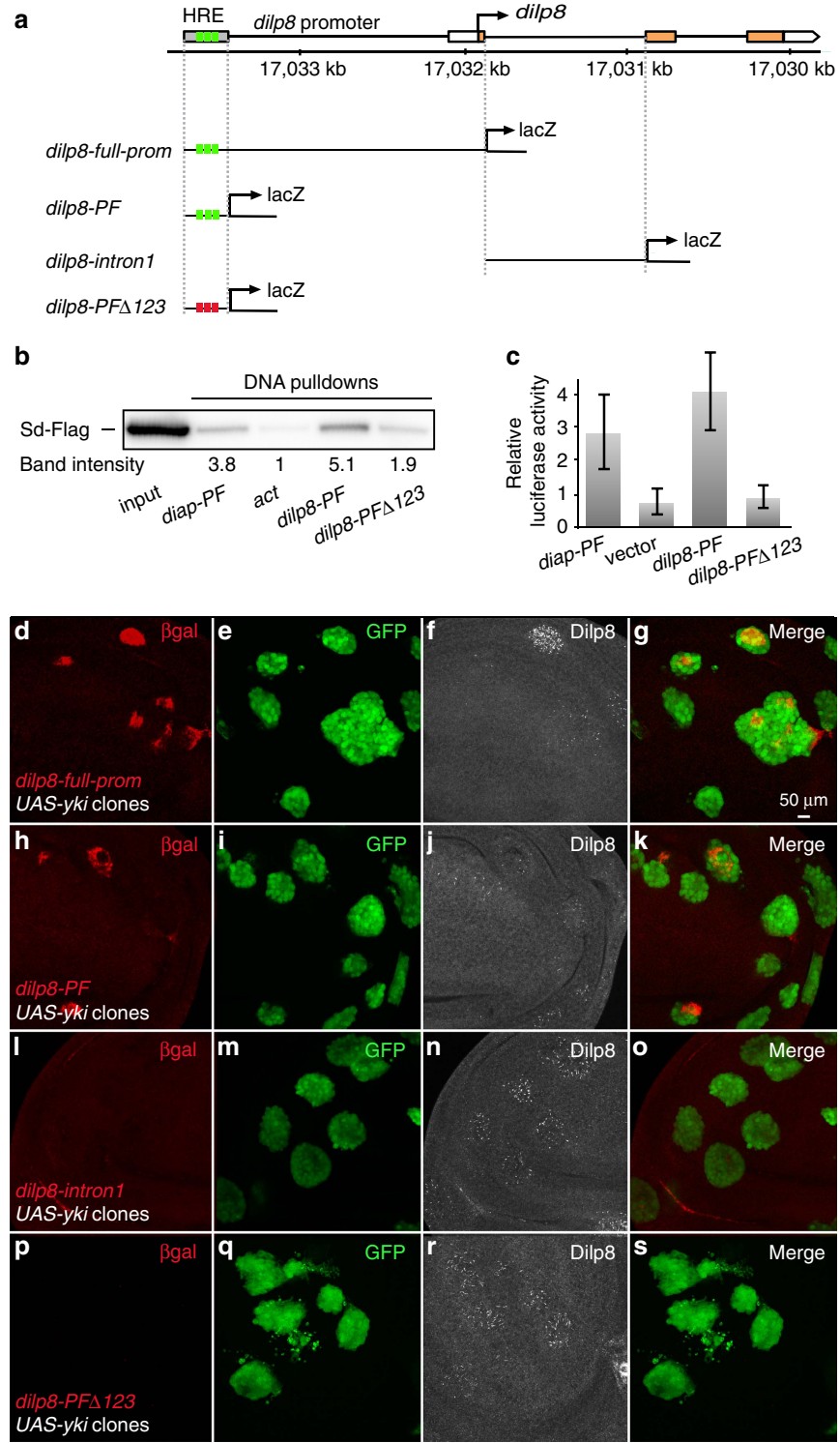

**Figure 3 | Yki directly regulates *dilp8* expression through a HRE in the *dilp8* promoter.** (**a**) Schematic of the *dilp8* promoter region and the HRE harbouring the three putative Sd-binding sites (indicated as green squares). The *dilp8* promoter fragments used to study Yki-dependent regulation of *dilp8* expression *in vivo* are shown. In *dilp8-PFΔ123*, mutations of the three putative Sd-binding sites are indicated as red squares. (**b**) DNA pull-down experiments show that binding of Sd to the *dilp8-PF* is mediated by the three Sd-binding sites. The indicated DNA fragments were incubated with lysates from S2 cells transfected with Sd-Flag. Band intensities represents the average of three independent experiments: for *diap-PF*: 3.8 ± 0.7 (positive control), for *act*: 1 ± 0.2 (negative control), for *dilp8-PF*: 5.1 ± 1.0, and for *dilp8-PF Δ123*: 1.9 ± 0.4. (**c**) Luciferase assay showing that Yki/Sd activate gene expression through the HRE in the *dilp8-PF*. S2 cells were transfected with Yki and Sd. The ability of Yki/Sd to induce gene expression from the indicated promoter fragments was measured (triplicate samples, error bars represent s.e.m.). (**d–s**) Yki induces *dilp8* transcription through the HRE *in vivo*. Wing imaginal discs carrying GFP-labelled Yki-expressing clones were dissected from transgenic flies carrying the indicated *dilp8* promoter fragments fused to the *LacZ*-encoding sequence. The full *dilp8* promoter and *dilp8-PF*, but not *dilp8-intron1* and *dilp8-PFΔ123*, induces *lacZ* expression as detected by β-gal staining (**d,h,l,p** in red) in the GFP-labelled *yki*-overexpressing clones (**e,i,m,n** in green). In each condition, *yki* overexpression leads to elevated levels of endogenous Dilp8 protein (**f,j,n,r** in white).

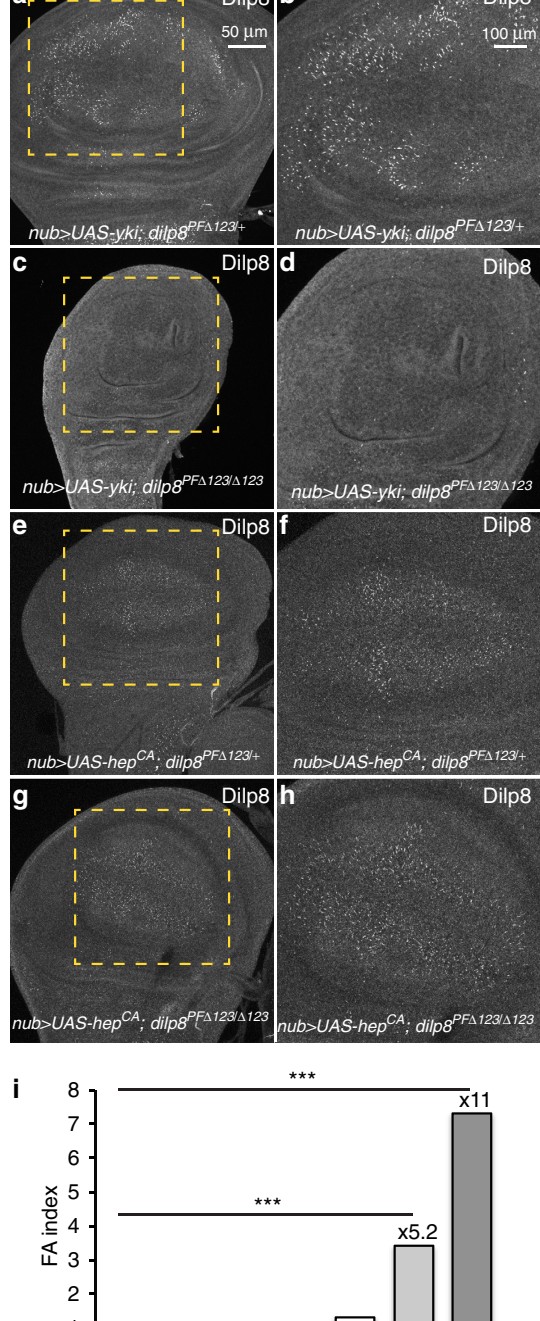

**Figure 4 | Yki-dependent regulation of Dilp8 adjusts organ growth and limits developmental instability.** (**a,c,e,g**) JNK-dependent regulation of *dilp8* expression is not affected in *dilp8-PFΔ123* mutants. (**b,d,f,h**) represent higher magnification images of the area shown in the dashed lines. Wing discs were dissected from the indicated genotypes and stained for Dilp8. (**i**) Mutation of the three Sd-binding sites in the *dilp8* promoter by gene editing induces FA. Bar histograms shows FA indices (FAi) of left and right wing areas measured on individuals of the indicated genotypes. ***$P < 0.0001$, $F$-test for unequal distributions.

anti- rat or Alexa Fluor 647 goat anti-rat, Cy3-conjugated donkey anti-chicken from Jackson ImmunoResearch) for 2 h at room temperature. After washing, tissues were mounted in Vectashield containing DAPI for staining of DNA (Vector Labs). Fluorescence images were acquired using a Leica SP5 DS ($\times$20 and $\times$40 objectives) and processed using Adobe Photoshop CS5 or Image J. Wing areas were measured using Image J.

**Antibodies.** The following primary antibodies were used, rat anti-Dilp8: 1/500 (ref. 6) and chicken anti-beta-galactosidase: 1/1,000 (GeneTek).

**Plasmids.** *dilp8*, *diap* and *actin* promoter sequences were PCR-amplified from genomic DNA (DNA preparations made using DNeasy Blood and Tissue kit (Qiagen)) and cloned in the pENTR/D-Topo vector (Invitrogen) using the following primers: for pENTR-dilp8-PF: sense, 5′-GTA TGATAGCAGGGTTCTG TG-3′ and antisense, 5′-AGG TTG TGA AAT TGT ATT TAT GAA TGA-3′, for pENTR-diap-PF: sense, 5′-ATT TGC CCT TGT CTG TAG TTG C-3′ and antisense, 5′-ATC GAGCAATGGATAAACCGTGAG-3′, for pENTR-act-PF: sense, 5′-GGACTC GTCGTACTCCTGCTTG-3′ and antisense, 5′-CCC ATC TAC GAG GGT TAT GC-3′. To generate the pENTR-dilp8-PF Δ123 plasmid, mutations were introduced in pENTR-dilp8-PF using the kit QuickChange site-directed mutagenesis kit (Stratagene) and the following primers: Dilp8 MutSd S1: 5′-TTC GCA CGT ATC CTT CTT CAT GTG CAT TCT CGC GCT CGT C-3′, Dilp8 MutSd A1: 5′-GAC GAG CGC GAG AAT GCA CAT GAA GAA GGA TAC GTG CGA A-3′, Dilp8 MutSd S3: 5′-CCA CTT GCT TTT CGC TTT CGT TGC ATG AAC GCA TGC GGA AAT GTT A-3′, Dilp8 mutsd A3: 5′-TAA CAT TTC CGC ATG CGT TCA TGC AAC GAA AGC GAA AAG CAA GTG G-3′, Dilp8 mut S4: 5′-ACA CCG ACT TGG GTT CGG ATC CTG TGC GCT GGC CGT T-3′, Dilp8 Mut A4: 5′-AAC GGC CAG CGC ACA GGA TCC GAA CCC AAG TCG GTG T-3′. To generate the beta-galactosidase reporter plasmids, sequences corresponding to the full *dilp8* promoter, intron I, dilp8-PF and dilp8-PF Δ123 were amplified from plasmids or genomic DNA, and cloned into the pCASPER-attbN-hs43-lacZ plasmid (kind gift from Ulrike Lohr and Herbert Jäckle, Max Planck Institut für Biophysikalische Chemie, Göttingen, Germany). The following primers were used: for intron I: sense, 5′-CTG CAA TGG AGG AGC GGG-3′ and antisense, 5′-GTG AGT TCG AGT TGA AGT TAA ACT-3′, for dilp8-PF and dilp8-PF Δ123: sense, 5′-GTA TGA TAG CAG GGT TCT GTG-3′ and antisense, 5′-AGG TTG TGA AAT TGT ATT TAT GAA TGA-3′, for the full *dilp8* promoter region: sense, 5′-TTT AGT GGC GTT CTA GGA-3′ and antisense, 5′-AGG TTG TGA AAT TGT ATT TAT GAA TGA-3′. To generate transgenic reporter lines, the pCASPER-attbN-hs43-lacZ constructs were introduced into the germ line by injections in the presence of the PhiC31 integrase and inserted in the landing site 51C1 on the 2rd chromosome (Bloomington Drosophila Stock Center, BL24482, BestGene Inc).

***dilp8* editing by accelerated homologous recombination.** To generate a *dilp8* null mutant, an approach combining the CRISPR technique and homologous recombination was used as described in ref. 24. Double-strand breaks were induced by the CRISPR technique using single-stranded guide (sg)RNAs and a Cas9-encoding plasmid. For optimal targeting of the *dilp8* locus, sgRNA target sequences were selected as 20-nt sequences preceding an NGG PAM sequence in the genome (GN20GG). To generate pCFD4{dilp8$^{KO}$}, gRNAs targeting exon I and exon II in the *dilp8* locus were cloned into the tandem gRNA expression vector, pCFD4 (kind gift from Simon Bullock (Addgene plasmid # 49411)), using the *dilp8*-5′-KO-pCFD4-FORWARD: 5′-TAT ATA GGA AAG ATA TCC GGG TGA ACT TCg cca cta aaa tga gtt caa GTT TTA GAG CTA GAA ATA GCA AG-3′ and *dilp8*-3′-KO-pCFD4-REVERSE: 5′-ATT TTA ACT TGC TAT TTC TAG CTC TAA AAC tca ggc aac aga ctc cga tgac GAC GTT AAA TTG AAA ATA GGT C-3′ (*dilp8*-specific sequences are in lower case) primers as described in http://www.crisprflydesign.org/wp-content/uploads/2014/06/Cloning-with-pCFD4.pdf. For homologous recombination, two homology arms were amplified from genomic DNA using the following primers: for homology arm I, sense: 5′-**GGT ACC** TGA TGG GCA GCT CGT CGG GCA GCT TC-3′ and antisense: 5′-**GCG GCC GC**g TAA ATG GAT CTG TGT CCC TGG GA-3′, for homology arm II, sense: 5′-**ACT AGT** AAC TCA TTT TAG TGG CGT TCT AGG ATC A-3′ and antisense: 5′-**AGA TCT** CAA ATG ATT ACC GCA TTA AAG CTA ATC AC-3′ (in bold are the added restrictions sites used for cloning into pTV2). The resulting PCR products were digested and cloned into the pTV2 vector (kind gift from Cyril Alexandre and Jean-Paul Vincent, the Francis Crick Institute) to generate pTV2{dilp8$^{KO}$}. To facilitate homologous recombination, embryos were injected with pTV2{dilp8$^{KO}$} in the presence of pCFD4{dilp8$^{KO}$} and a CAS9-containing plasmid. To confirm gene targeting the following primers were used: *dilp8KO* S: 5′-GGA CGG GTT AAC CAT TCA GCA AGT TAG-3′ and *dilp8KO Rev*: 5′-CGC GAA CTC GAT GGA CCT TCT GTC GC-3′. After confirmed targeting, the majority of the targeting vector was removed by crossing to a strain expressing a constitutively active Cre (Bloomington stock BL1092; procedure outlined in ref. 24. The resulting strain is referred to as *dilp8$^{KO}$* in the manuscript.

To generate a *dilp8* mutant deleting a 600 bp fragment of the promoter encompassing the three scalloped-binding sites (*dilp8$^{pTV2ΔPF}$*), gRNAs directed against sequences upstream and downstream of the scalloped-binding sites were

cloned into pCFD4 as described above using the following primers: *dilp8 HR* S: 5′-TAT ATA GGA AAG ATA TCC GGG TGA ACT TCg aac ttt cac tgt cgt ttc tgt tGT TTT AGA GCT AGA AAT AGC AAG-3′ and *dilp8 HR REV*: 5′-ATT TTA ACT TGC TAT TTC TAG CTC TAA AAC gtg gcg cga ttc ttt aga tcG ACG TTA AAT TGA AAA GAA GTC-3′. The primers used to generate the homology arms for the pTV2 vector were: HRR S: 5′-**GCG GCC GCG GC**G GAC TCA CTG AGT CAC AGT CAC ATG-3′, HRR AS: 5′-**GGT ACC** AGA AAC GAC AGT GAA AGT TCT ACACTT TCG-3′, HRL S: 5′-**ACT AGT** CTA AAG AAT CGC GCC ACT TG-3′, HRL AS: 5′-**AGA TCT** CAG ACA TCT TAG TCA TGG TCT G-3′ (in bold are the added restrictions sites used for cloning into pTV2). To facilitate homologous recombination, embryos were injected with pTV2{dilp8-ΔPF} in the presence of pCFD4{dilp8-ΔPF} and a CAS9-encoding plasmid. The resulting strain, referred to as *dilp8^{pTV2ΔPF}*, was used as a host for reintegration of the missing 600 bp promoter fragment harbouring point mutations in the three scalloped-binding sites (dilp8-PF Δ123) or not (dilp8-PF, control) via the attP site. For this purpose, dilp8-PF and dilp8-PFΔ123 were PCR-amplified from plasmids and cloned into the reintegration vector RIV^{cherry} (kind gift from Cyrille Alexandre and Jean-Paul Vincent, the Francis Crick Institute) giving rise to RIV^{cherry}{dilp8-PF Δ123, mini-white} and RIV^{cherry}{dilp8-PF, mini-white}. For the PCR amplification of dilp8-PF and dilp8-PF Δ123 the following primers were used: RIV S: 5′-**ACT AGT** GTT CGG CTC TGC CTC TGT CTC TTT CGG-3′ and RIV AS: 5′-**AGA TCT** ATC AGG TTG TGA AAT TGT ATT TAT GAA T-3′ (in bold are the added restrictions sites used for cloning into RIV^{cherry}). Reintegration was achieved by injecting a *dilp8^{pTV2}* strain expressing the PhiC31 integrase. After confirmed targeting of the reintegration vector into the *dilp8* locus, the majority of the targeting vector was removed by crossing to a strain expressing a constitutively active Cre (Bloomington stock BL1092; procedure outlined in ref. 24). Sequencing analysis of the *dilp8-PFΔ123* mutant were performed using the following primers: *Dilp8 HRE* S2: 5′-GGT AGG GGC ATT CGA CGG AGA TCG TTG-3′ and *Dilp8 HRE* Rev2: 5′-GGA AGG CCA GCG AAA TTG TTG TTA AAC-3′.

**Luciferase assays.** *dilp8*, *diap* and *actin* promoter sequences were PCR-amplified and cloned into pGL3basic vector (Promega) using the kit Dual Luciferase Reporter Assay System (Promega). For luciferase assays, S2R+ cells were transfected for 3 days with the following vectors: *Firefly-luciferase pGL3-promoter* (500 ng), *pAct-Renilla-luciferase* (100 ng,), pAct-yki (600 ng), pAct-sd (400 ng). All experiments were performed in triplicate, and luciferase activities were normalized against *Renilla* luciferase activity following indications of the Dual Luciferase Reporter Assay System protocol (Promega).

**DNA pull-down.** The method was adapted from ref. 32. *Drosophila* S2 cells were plated in a six-well plate at a density of $2 \times 10^6$ cells per well. Each well was transfected with 400 ng of pAC Flag-Sd (Kind gift from Clara Sidor and Barry Thompson, the Francis Crick Institute). In all, 48 h later, cells were washed once in PBS and lysed in 500 µl IP buffer (50 mM Tris pH7.5, 150 mM NaCl, 1% Triton-X-100, 1 mM EGTA) in the presence of protease (complete mini tablets from Roche) and phosphatase (phosphatase inhibitor cocktail 1, Sigma) inhibitors for 10 min on ice. Cell extracts were then spun down at 13,000 r.p.m. for 10 min at 4 °C to remove cell debris. Cells extracts were incubated for 1 h at 4 °C on a rotating wheel with a short non-biotinylated actin probe to pre-clear the extracts from non-specific DNA-binding proteins (10 µg probe per 100 µg protein). In all, 30 µl Neutravidin beads slurry (Thermoscientific) were washed in IP buffer and incubated with biotinylated probes (Dilp8-PF, Dilp8-PF Δ123, DIAP1 or Act5) for 1 h at 4 °C on a rotating wheel. Pre-cleared cell extracts and probe-bound beads were combined and incubated overnight at 4 °C on a rotating wheel. Beads were then washed $3 \times 5$ min in IP buffer and resuspended in 30 µl SDS loading buffer for western blot analysis. DNA probes were amplified from genomic DNA using the following primers: for the DIAP1 probe (583 bp) F: 5′-ATT TGC CCT TGT CTG TAG TTG C-3′ and R: 5′-[Btn]ATC GAG CAA TGG ATA AAC CGT GAG-3′, for the Act5C probe (606 bp) F: 5′-GGA CTC GTC GTA CTC CTG CTT G-3′ and R: 5′-[Btn]CCC ATC TAC GAG GGT TAT GC-3′, for the Dilp8-PF probe (680 bp) F: 5′-[Btn]GTA TGA TAG CAG GGT TCT GTG-3′ and R: 5′-AGG TTG TGA AAT TGT ATT TAT GAA TGA-3′, and for the Dilp8-PF Δ123 probe (680 bp) F: 5′-[Btn]GTA TGA TAG CAG GGT TCT GTG-3′ and R: 5′-AGG TTG TGA AAT TGT ATT TAT GAA TGA-3′ (all probes were 5′ biotinylated). Short *actin* probe for cell extract pre-clearing (60 bp) F and R oligonucleotides were annealed at a final concentration of 1 µg µl⁻¹ double-stranded DNA: F: 5′-GAG CAC GGT ATC GTG ACC AAC TGG GAC GAT ATG GAG AAGATC TGG CAC CAC ACC TTC TAC-3′ and R: 5′-GTA GAA GGT GTG GTG CCA GAT CTT CTC CAT ATC GTC CCA GTT GGT CAC GAT ACC GTG CTC-3′. See uncropped image of the western blotting experiment in Supplementary Fig. 1.

**Quantitative RT-PCR.** Larvae were collected at the indicated number of hours AED. Whole larvae or dissected larval wing discs were frozen in liquid nitrogen. Total RNA was extracted from whole larvae or dissected tissues using a QIAGEN RNeasy lipid tissue minikit or microkit according to the manufacturer's protocol. RNA samples (2 µg per reaction) were treated with DNase and reverse-transcribed using SuperScript II reverse transcriptase (Invitrogen), and the generated cDNAs were used for real-time RT-PCR (StepOne Plus; Applied Biosystems) using

PowerSYBRGreen PCR mastermix (Applied Biosystems), with 8 ng of cDNA template and a primer concentration of 300 nM. Samples were normalized to levels of ribosomal protein (rp)49 transcript levels. Three separate biological samples were collected for each experiment and triplicate measurements were performed. Primers were designed using PrimerExpress software (Applied Biosystems) as follows: dilp8S 5′-CGA CAG AAG GTC CAT CGA GT-3′, dilp8R 5′-GTT TTG CCG GAT CCA AGT C-3′.

**Western blotting.** Proteins were resolved by SDS-PAGE using 12% gels (NuPAGE Novex gel, Invitrogen) and transferred electrophoretically to polyvinylidene difluoride membranes (Amersham). The membranes were incubated for 1 h in blocking buffer (PBS, 5% milk) and incubated overnight at 4 °C in the same buffer containing primary antibodies at 1:1,000 dilutions (mouse anti-Flag F3165 (Sigma)). Membranes were washed three times in PBS-T, blocked for 1 h, and probed with secondary antibodies in blocking buffer for 1 h at room temperature. After three washes in PBS-T, chemiluminescence was observed using the ECL-Plus western blotting detection system (Amersham Biosciences). Images were generated using the Fujifilm Multi Gauge software. The uncropped western blot (for Fig. 3) can be found in Supplementary Fig. 1.

**Pupariation curves.** L1 larvae were collected 24-h AED on agar plates with yeast (4-h egg collections) and reared in tubes (thirty larvae each) containing standard food (see above). The number of larvae that had pupariated at a given time AED was scored every 6 h.

**Measurement of the FA index.** L1 larvae were collected 24 h AED (4-h egg collections) and reared at 30 animals per tube at 26.5 °C. Adult flies of the appropriate genotypes were collected, stored in ethanol and mounted in an Euparal solution. Pictures of dissected wings were acquired using a Leica Fluorescence Stereomicroscope MZ16 FA with a Leica digital camera DFC 490. Wing areas were measured using Image J. We used the FA index to assess intra-individual size variation between left and right wing areas as described in ref. 12. FA index $= \Sigma(Ai)^2/n$, where Ai are the normalized differences between left and right wing area within a given individual ($n > 22$). P values are the results of a F test provided by Microsoft Excel.

**Statistics.** P values are the results of a Student's test provided by Microsoft Excel (*P < 0.05; **P < 0.01).

**Genotypes.** In Fig. 1a: $w^-$; *elav-Gal80/+*; *rn-Gal4/+*, $w^-$; *elav-Gal80/+*; *rn-Gal4, UAS avl RNAi GD/+*, $w^-$; *elav-Gal80/rh3 RNAi KK*; *rn-Gal4, UAS avl RNAi GD/+*, $w^-$; *elav-Gal80/sd RNAi KK*; *rn-Gal4, UAS avl RNAi GD/+*, $w^-$; *elav-Gal80/yki RNAi KK*; *rn-Gal4, UAS avl RNAi GD/+*. Fig. 1b: $w^-$; *elav-Gal80/+*; *rn-Gal4/+*, $w^-$; *elav-Gal80/+*; *rn-Gal4, UAS avl RNAi GD/+*, $w^-$; *elav-Gal80/sd RNAi KK*; *rn-Gal4, UAS avl RNAi GD/+*, $w^-$; *elav-Gal80/sd RNAi KK*; *rn-Gal4, UAS avl RNAi GD/+*. Fig. 1c,d: $w^-$; *spalt-Gal4/+*; +. Fig. 1e,f: $w^-$; *spalt-Gal4/+*; *UAS-yki/+*. Fig. 1g,h: $yw^-$; *hep^{75}*; *spalt-Gal4/+*; *UAS-yki/+ (male)*. Fig. 1i: $w^-$; *spalt-Gal4/+*; +, $w^-$; *spalt-Gal4/+*; *UAS-yki/+*, $yw^-$; *hep^{75}*; *spalt-Gal4/+*; *UAS-yki/+ (male)*.

In Fig. 2a,d: $yw^-$, *hs-FLP*; *Act > CD2 > RFP/+*; *dilp8-GFP/UAS-yki*. Fig. 2e,f: $yw^-$, *hs-FLP*; *Act > CD2 > GFP/+*; +. Fig. 2g,h: $yw^-$, *hs-FLP*; *Act > CD2 > GFP/+*; *UAS-yki/+*. Fig. 2i,f: $yw^-$, *hs-FLP*; *Act > CD2 > GFP/sd RNAi KK*; *UAS-yki/+*. Fig. 2k,l: $yw^-$, *hs-FLP*; +; *FRT82B, ex^{e1}/FRT82B, ubi-GFP*. Fig. 2m,n: $yw^-$, *hs-FLP*; *FRT42D, hpo^{5-1}/FRT42D, ubi-GFP*; +. Fig. 2o,p: $yw^-$, *hs-FLP*; +; *FRT82B, LATS^{X1}/FRT82B, ubi-GFP*.

In Fig. 3d,g: $yw^-$, *hs-FLP*; *Act > CD2 > GFP/dilp8-full-promoter-LacZ*; *UAS-yki/+*. Fig. 3h,k: $yw^-$, *hs-FLP*; *Act > CD2 > GFP/dilp8-PF-LacZ*; *UAS-yki/+*. Fig. 3l,o: $yw^-$, *hs-FLP*; *Act > CD2 > GFP/dilp8-intron1-LacZ*; *UAS-yki/+*. Fig. 3p,s: $yw^-$, *hs-FLP*; *Act > CD2 > GFP/dilp8-PFΔ123-LacZ*; *UAS-yki/+*.

In Fig. 4a,b: $w^-$; *nub-Gal4/UAS-yki::GFP*; *dilp8^{PFΔ123}/+*. Fig. 4c,d: $w^-$; *nub-Gal4/UAS-yki::GFP*; *dilp8^{PFΔ123}/dilp8^{PFΔ123}*. Fig. 4e,f: $w^-$; *nub-Gal4/ UAS-hep^{CA}*; *dilp8^{PFΔ123}/+*. Fig. 4g,h: $w^-$; *nub-Gal4/UAS-hep^{CA}*; *dilp8^{PFΔ123}/ dilp8^{PFΔ123}*. Fig. 4i: $w^-$; +; +, $w^-$; +; *dilp8^{PF123}/+*, $w^-$; +; *dilp8^{PFΔ123}/+*, $w^-$; +; *dilp8^{KO}/+*, $w^-$; +; *dilp8^{PF123}/dilp8^{KO}*, $w^-$; +; *dilp8^{PFΔ123}/ dilp8^{KO}*, $w^-$; +; *dilp8^{KO}/dilp8^{KO}*.

**Data availability.** All data are available in the article or its Supplementary Files or available from the authors on request.

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

## Acknowledgements

We thank Gisele Jarretou and Alessandra Mauri for technical assistance, and V. Virolle for help with the CRISPR/Cas9 methodology. We thank the Vienna *Drosophila* RNAi Center, the *Drosophila* Genetics Resource Center and the Bloomington stock center for providing *Drosophila* lines. This work was supported by the CNRS, INSERM, European Research Council (Advanced grant no. 268813 to P.L.), ARC (grant PGA120150202355), FRM (funding to E.B.) and the Labex Signalife program (grant ANR-11-LABX-0028-01 to P.L.).

## Author contributions

Conceptualization, methodology and writing, E.B., J.C., D.S.A. and P.L. Investigations, E.B., J.C. and D.S.A. Funding acquisition and supervision, P.L.

## Additional information

**Competing financial interests:** The authors declare no competing financial interests.

