## [Peer Review File · Nature Communications]

Reviewer #1 (Remarks to the Author)

A. This manuscript identifies Yki as a regulator of Dilp8 expression, and shows that direct regulation of Dilp8 by Yki contributes to the consistent regulation of organ size in *Drosophila*.

B. This discovery addresses a fundamental gap in our understanding of how the growth of individual organs is coordinated with developmental transitions and the growth of other organs. It also helps to explain why many overgrowth mutants in the fly delay pupariation. It's a short paper, but the analysis is compelling, the results are important, and I think it will be of widespread interest.

C. Fine

D. Fine

E. Fine

F. One suggestion: It could be helpful to have some sample pictures of representative wings to go along with the FA analysis in Fig 4e (even in a supplementary figure), as FA index numbers are not intuitive for most readers, and it isn't even properly explained in the methods (readers are referred to another paper).

G. Fine

H. Fine

Reviewer #2 (Remarks to the Author)

This is an important and interesting manuscript. The authors, in my opinion, provide the first convincing evidence for a mechanism by which the Hippo pathway could regulate growth and size during the normal development of an organism. The experiments are clear, straightforward and presented in a logical sequence.

This manuscript should be published in Nature Communications promptly. I have some suggestions that would improve the accessibility of the manuscript:

1) The authors use the term "fluctuating asymmetry" in the Introductory Paragraph and Introduction without really explaining it. Given the importance of this concept in analyzing the HRE mutant phenotype, the authors need to explain it clearly.

2) Figure 4e shows the increase in fluctuating asymmetry elicited by mutating the HREs in the dilp8 promoter. A reader could go away thinking that the primary role of this mechanism is to match the size of the two wings. The same phenotype could also be obtained simply by weakening the size control mechanism of each wing disc independently without having any mechanism for size matching. I would like to see a supplementary figure or table that shows the overall distribution of wing sizes in each population (control and delta-HRE). I would like to know if the overall distribution of wing sizes is wider in one population when compared to the other. I'm sure the authors already have these data.

Minor points:

1) The description of the genome editing experiments is a little hard to follow. A supplementary figure would help.

2) Please state in the figure legend that panels 1c'-e' represent higher magnification images of the area shown in the dashed lines. The same for 4c' and d'.

Reviewer #1 (Remarks to the Author):

A. This manuscript identifies Yki as a regulator of Dilp8 expression, and shows that direct regulation of Dilp8 by Yki contributes to the consistent regulation of organ size in *Drosophila*.

B. This discovery addresses a fundamental gap in our understanding of how the growth of individual organs is coordinated with developmental transitions and the growth of other organs. It also helps to explain why many overgrowth mutants in the fly delay pupariation. It's a short paper, but the analysis is compelling, the results are important, and I think it will be of widespread interest.

C. Fine

D. Fine

E. Fine

F. One suggestion: It could be helpful to have some sample pictures of representative wings to go along with the FA analysis in Fig 4e (even in a supplementary figure), as FA index numbers are not intuitive for most readers, and it isn't even properly explained in the methods (readers are referred to another paper).

As requested, we have added a new figure with representative superimposed pictures of left/right wings for each genotypes (see new Suppl. Fig. S3).

G. Fine

H. Fine

Reviewer #2 (Remarks to the Author):

This is an important and interesting manuscript. The authors, in my opinion, provide the first convincing evidence for a mechanism by which the Hippo pathway could regulate growth and size during the normal development of an organism. The experiments are clear, straightforward and presented in a logical sequence.

This manuscript should be published in Nature Communications promptly. I have some suggestions that would improve the accessibility of the manuscript:

1) The authors use the term "fluctuating asymmetry" in the introductory Paragraph and Introduction without really explaining it. Given the importance of this concept in analyzing the HRE mutant phenotype, the authors need to explain it clearly.

As requested, we have added a short description of fluctuating asymmetry (FA) in the introductory paragraph.

2) Figure 4e shows the increase in fluctuating asymmetry elicited by mutating the HREs in the *dilp8* promoter. A reader could go away thinking that the primary role of this mechanism is to match the size of the two wings. The same phenotype could also be obtained simply by weakening the size control mechanism of each wing disc independently without having any mechanism for size matching. I would like to see a supplementary figure or table that shows the overall distribution of wing sizes in each population (control and delta-HRE). I would like to know if the overall distribution of wing sizes is wider in one population when compared to the other. I'm sure the authors already have these data.

The Reviewer makes a good point. We now present the overall distribution of wing sizes in the different genotypes in new Suppl. Fig. S4. As shown in this figure, the wing size distribution is not modified in the *PF1123/KO* animals compared to controls, although FA is strongly increased. This suggests that the FA phenotype in this genetic background is not due to a general impairment of the size control mechanism, but rather to a loss of « size matching ». We have added a comment concerning this new data in the main text.

Minor points:

1) The description of the genome editing experiments is a little hard to follow. A supplementary figure would help.

We have added a scheme illustrating the gene editing method, now presented in Suppl. Fig. S1.

2) Please state in the figure legend that panels 1c'-e' represent higher magnification images of the area shown in the dashed lines. The same for 4c' and d'.

We thank the Reviewer for pointing these missing descriptions to us and have modified the legend of Fig. 1 and 4 accordingly.